# Bluebelle pilot randomised controlled trial of three wound dressing strategies to reduce surgical site infection in primary surgical wounds

Jane Blazeby, On behalf of the Bluebelle Study Group

Department of Population Health Sciences, Bristol Medical School, University of Bristol, Bristol, UK

**Correspondence to**
Professor Jane Blazeby;
j.m.blazeby@bristol.ac.uk

## ABSTRACT

**Objective** Surgical site infection (SSI) affects up to 25% of primary surgical wounds. Dressing strategies may influence SSI risk. The Bluebelle study assessed the feasibility of a multicentre randomised controlled trial (RCT) to evaluate the effectiveness and cost-effectiveness of different dressing strategies to reduce SSI in primary surgical wounds.

**Design** A pilot, factorial RCT.

**Setting** Five UK hospitals.

**Participants** Adults undergoing abdominal surgery with a primary surgical wound.

**Interventions** Participants were randomised to 'simple dressing', 'glue-as-a-dressing' or 'no dressing', and to the time at which the treatment allocation was disclosed to the surgeon (disclosure time, before or after wound closure).

**Primary and secondary outcome measures** Feasibility outcomes focused on recruitment, adherence to randomised allocations, reference assessment of SSI and response rates to participant-completed and observer-completed questionnaires to assess SSI (proposed primary outcome for main trial), wound experience and symptoms, and quality of life (EQ-5D-5L).

**Results** Between March and November 2016, 1115 patients were screened; 699 (73.4%) were eligible and approached, 415 (59.4%) consented and 394 (35.3%) were randomised (simple dressing=133, glue=129 and 'no dressing'=132). Non-adherence to dressing allocation was 2% (3/133), 6% (8/129) and 15% (20/132), respectively. Adherence to disclosure time was 99% and 86% before and after wound closure, respectively. The overall rate of SSI (reference assessment) was 18.1% (51/281). Response rates to the Wound Healing Questionnaire and other questionnaires ranged from >90% at 4 days to 68% at 4–8 weeks.

**Conclusions** A definitive RCT of dressing strategies including 'no dressing' is feasible. Further work is needed to optimise questionnaire response rates.

**Trial registration number** 49328913; Pre-results.

## Strengths and limitations of this study

► This pilot factorial randomised controlled trial addressed whether a main trial of wound dressing strategies was possible.
► The factorial design examined whether intraoperative randomisation was acceptable and feasible.
► Surgical trainee collaboratives and research nurse teams worked together to optimise recruitment.
► Temporary skin transfers adjacent to the surgical wound supported adherence to dressing allocation.
► Follow-up questionnaire response rates were low and need optimisation in a main trial.

minimise infection, many develop a surgical site infection (SSI). This is especially problematic in abdominal surgery and high-risk settings where rates of SSI may reach 25%.[2 3] SSIs require antibiotics and multiple dressings, can delay recovery, reduce quality of life and are expensive for health services.[4 5]

Abdominal surgery carries one of the highest rates of SSI, particularly if the operation involves the colon or rectum.[3 6] Caesarean section is another procedure which carries a high rate of SSI.[7] Possible ways to reduce SSI include modification of preoperative, perioperative and postoperative factors, which include optimising wound dressing strategies and examining whether dressings are needed at all. A Cochrane review of randomised controlled trials (RCTs) examining different dressing strategies, which included studies of wounds without a dressing, was performed in 2011 and since updated.[8–10] The initial review found no difference in rates of SSI between wounds covered with different dressings or left uncovered. The update found insufficient evidence to reach a firm conclusion. Most trials included in the review were small and at high or unclear risk of bias. A subsequent Cochrane review of intraoperative methods to reduce SSI commented on

## INTRODUCTION

Each year, there are over 5 million hospital admissions for surgery in England alone.[1] The majority result in 'a closed primary wound' and it is common practice to cover these with a dressing. Despite attempts to

the need for more research in this field.[11] In 2014, the UK National Institute of Health Research, therefore, called for research proposals to address these issues with feasibility and pilot work to establish if a major RCT was possible. The Bluebelle study, a programme of research including non-randomised feasibility projects (phase A) and a pilot RCT (phase B), was designed to inform the design of a main trial.[12–16] Phase A included interviews with key stakeholders to explore their views of dressings and a trial design,[12] a survey of surgical wounds to examine current dressing practice[13] and developmental work to design questionnaires to assess SSI[15] and other aspects of wound management.[14] Here, we report the pilot RCT.[16] The aim of the pilot RCT was to establish whether it would be feasible to carry out a large definitive RCT to compare the effectiveness and cost-effectiveness of different dressing strategies to reduce SSIs following elective and unplanned surgery with a primary wound. Specific objectives were to establish if it was possible to recruit and randomise, to assess the acceptability of, and adherence to, the trial interventions, to examine the feasibility of collecting follow-up data and to establish the measurement properties of the SSI Wound Healing Questionnaire (WHQ).

## METHODS

### Study design

A factorial design was used to investigate adherence to the allocated dressing types and the feasibility of randomising after wound closure. Patients were randomised 1:1:1 to dressing type (simple dressing, glue-as-a-dressing and 'no dressing') and 1:1 to the time at which the dressing allocation was disclosed to the surgeon (revealed before or after wound closure details had been entered onto the study database). Full details of the interventions are described in the protocol.[16] The randomisation sequences were generated by computer in advance of starting to recruit. Allocation was concealed until participant's eligibility and consent were documented and it was obtained via the internet. Depending on the randomisation result, the dressing allocation was either disclosed immediately, or the user was advised to log back on to the website after the wound had been closed. At the second log-on the user was asked to record the timing of wound closure, then the allocation was disclosed. The full protocol is published elsewhere.[16] The randomisation scheme was stratified by hospital and specialty (abdominal/obstetric surgery). The rationale for randomising to disclosure time as well as to dressing type was the need to understand whether surgeons' knowledge of treatment allocation influences the quality of wound closure (i.e., if allocation to 'no dressing' leads to surgeons taking more care with wound closure). It was intended to use in-theatre wound photography to assess quality of wound closure in relation to timing of disclosure of allocation; however, it soon became apparent that this outcome measure could not be implemented due to multiple governance and logistical

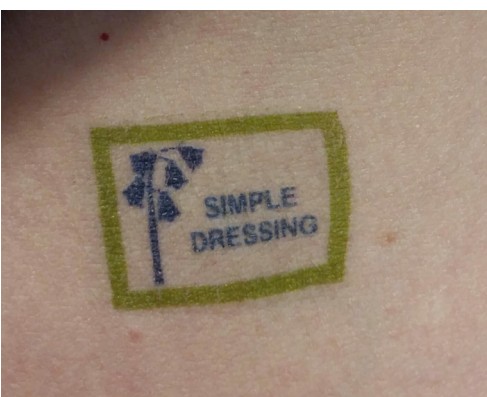

**Figure 1** Example of a skin transfer (modelled by a volunteer) that was applied near to the wound(s) to promote adherence to the dressing allocation.

challenges. This paper, therefore, reports the feasibility of conducting the pilot RCT of different dressing strategies and the feasibility of randomising before or after wound closure.

### Study setting and population

The study was set in University Hospitals Bristol NHS Foundation Trust (Bristol Royal Infirmary and St Michael's Hospital), North Bristol NHS Trust (Southmead Hospital), University Hospitals Birmingham NHS Foundation Trust (Queen Elizabeth Hospital) and Worcestershire Acute Hospitals NHS Trust. Included were adult participants undergoing abdominal general or obstetric surgery with a skin incision, who were able and willing to provide consent and complete follow-up at 4–8 weeks. Excluded were people who had undergone major surgery within the previous 3 months, wounds that a surgeon planned to close with tissue glue, contraindications to dressing allocation and prisoners. Surgery and wound closure were carried out according to local practice.

### Feasibility outcomes

Primary feasibility outcomes were whether patients were eligible, consented and recruited to the study, and whether they adhered to randomised allocation (yes/no). Skin transfers (temporary adherent tattoos) were placed next to the wound to encourage adherence to allocated dressing type after leaving the operating theatre (figure 1). The feasibility of collecting other data (likely to be used in a main trial) and their completeness was investigated for: patient and observer reported questionnaires measuring SSI with the newly validated WHQ; patient and observer reported questionnaires to assess symptoms and experiences of wounds and dressings; preference-based health-related quality of life (EuroQoL-five-dimension-five-level: EQ-5D-5L)[17]; wound complications and resource use.[18] A face-to-face wound assessment was carried out at 4-6 weeks to validate the WHQ.[19] This assessment was used in combination with data collected at discharge to classify each participant as having had an SSI or not.

## Sample size

It was calculated that 920 eligible participants would allow a consent rate of 36% (target number randomised=330) to be estimated with a 95% CI of 32% to 39%, and a recruitment rate of 60% with 95% CI of 56% to 64%. A consent rate of 36% was proposed because of previous experience recruiting into surgical trials. It was prespecified that, if adherence to dressing type was <70% in any group, it would be concluded that the main trial would not be feasible.

## Statistical analyses

Analyses were directed by a prespecified analysis plan and performed on an intention-to-treat basis. Continuous data were summarised as medians and IQRs. Categorical data were summarised as numbers and percentages and 95% CIs. Results were described by centre and by specialty as well as overall. The primary analysis took place when follow-up was complete for all recruited participants. All analyses were performed in Stata V.14.0 (StataCorp).

## Understanding adherence and acceptability to treatment allocation

Semistructured interviews were conducted with patients and staff within 30 days of surgery to understand issues relating to adherence and acceptability of dressing strategies (especially 'no dressing'). The findings have been reported elsewhere.[12]

## Patient and public involvement

Patients and the public were involved in several stages of this research. The initial idea came from a patient case study. A Bluebelle study patient and public involvement group was established including patients and their carers. Members were involved in study design and set up including commenting on patient facing materials. Patient representatives were on the study steering committee and management group and advised on how to approach patients and ideas for blinding study personnel. Extensive pretrial feasibility work (published) examined the burden of the intervention and time required to participate in the research with qualitative research. The main trial will continue to include patients throughout all of its stages (design, delivery, analyses, reporting and implementation).

# RESULTS

## Recruitment and participant details

Between March and November 2016, 1115 patients were screened; 699 (73.4%) were eligible and approached; 415 (37.2%) consented to take part; 394 (35.5%) were randomised (figure 2). The analysis population consisted of 388 participants (790 wounds), that is, the 394 randomised participants excluding three participants who withdrew and were unhappy for their data to be used, two participants who were allocated to disclosure of dressing allocation after wound closure and whose randomisation in theatre was not completed, and one participant whose surgery was cancelled. Some patients were consented but not randomised after consent because the study ended. Feasibility outcomes by centre are shown in table 1. Participants were predominantly women (227/388, 58.5%), overweight (median body mass index 28, IQR 24.3–31.6), American Society of Anesthesia grade 2 (203/384, 52.9%) and Caucasian 341/374 (91.2%) (table 2). Most wounds (93.7%) were closed with sutures and approximately three-quarters of participants were prescribed prophylactic antibiotics. There was no indication that these cointerventions were used differentially by group (table 2).

## Adherence to allocated treatment and timing of randomisation

Adherence to treatment allocation was good. More than 97% of participants correctly received the allocated dressing in theatre with adherence after leaving theatre to group allocation remaining high (86%) through to study exit. Adherence to the time at which their surgeons were informed about the treatment allocation was 99% and 86% before and after wound closure, respectively. Interviews with staff and patients indicated that skin transfers were acceptable; nobody objected to their use and most nurses viewed them as useful, although some felt they did not personally need to use the transfers as adherence aids.

## Follow up data

Face-to-face SSI reference assessments were performed in 80% of participants, among whom the overall SSI rate was 18.1% (table 3). Response rates for the participant and observer completed measures of SSI (WHQ) were 256/378 (68%) and 286/377 (76%), respectively, at 4–8 weeks (table 4). Completion of in-hospital questionnaires to assess Wound Symptoms and Experiences was >90% (355/385). Completion of EQ-5D-5L questionnaires during the follow-up was 269/382 (70%) at 15 days and 242/377 (64%) at 4–8 weeks. Wound complication data (other than SSI) were completed for 326/388 (84%) participants during the postoperative hospital stay and for 315/378 (83%) participants at 4–8 weeks, with similar completion rates for the three groups. Questionnaires documenting resource use during the admission for surgery were generally well completed (details not shown).

# DISCUSSION

Almost two-thirds of eligible patients consented to take part and adherence to allocated dressing type was good immediately after wound closure and during participants' follow-up. Therefore, it is concluded that a main trial of 'simple dressings', 'glue-as-a-dressing' and 'no dressings' is feasible and acceptable to patients and health professionals. Implementation of the different randomisation schedules (before or after wound closure) was generally successful. Reference SSI assessments were performed

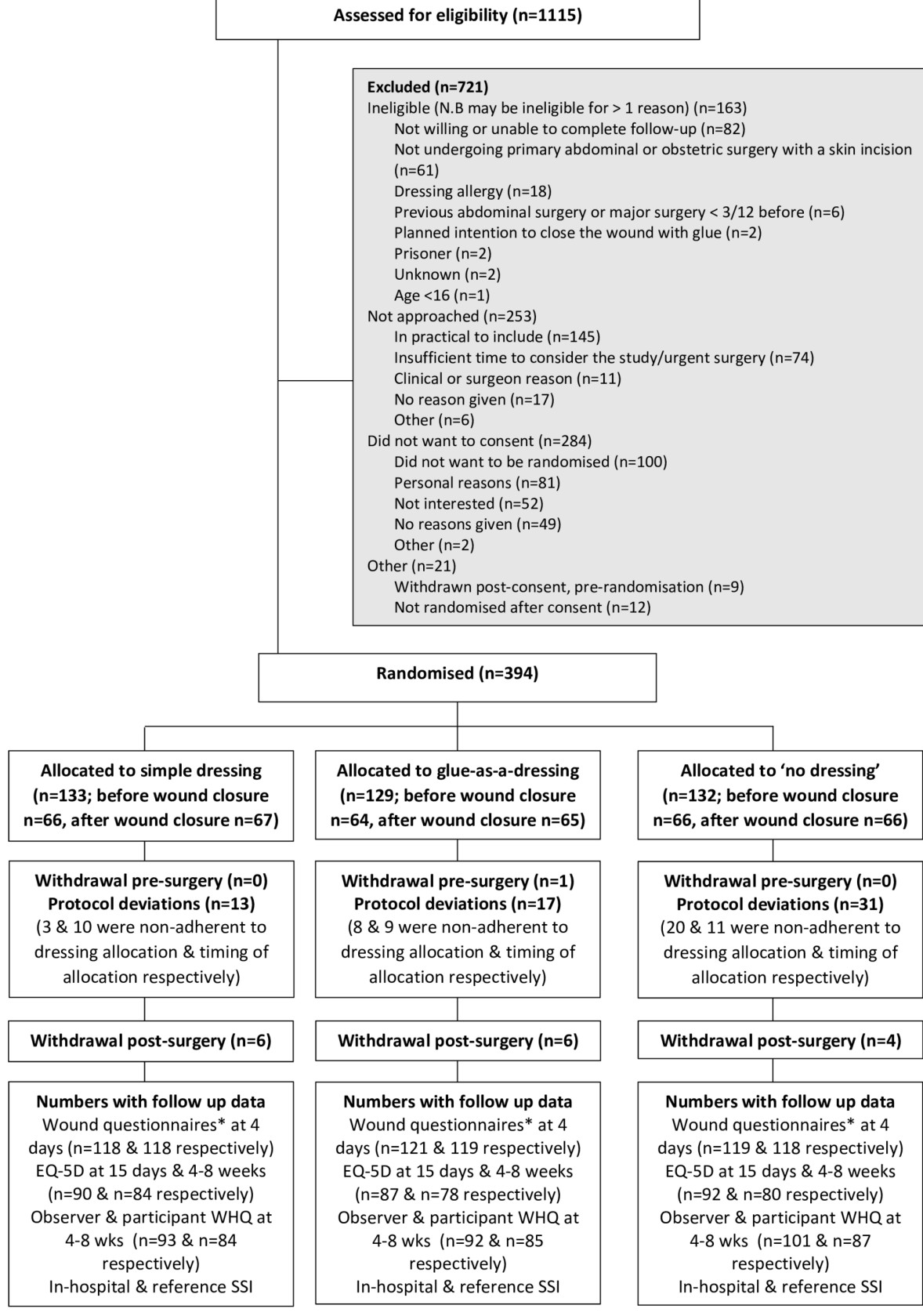

**Assessed for eligibility (n=1115)**

**Excluded (n=721)**
Ineligible (N.B may be ineligible for > 1 reason) (n=163)
 Not willing or unable to complete follow-up (n=82)
 Not undergoing primary abdominal or obstetric surgery with a skin incision
 (n=61)
 Dressing allergy (n=18)
 Previous abdominal surgery or major surgery < 3/12 before (n=6)
 Planned intention to close the wound with glue (n=2)
 Prisoner (n=2)
 Unknown (n=2)
 Age <16 (n=1)
Not approached (n=253)
 In practical to include (n=145)
 Insufficient time to consider the study/urgent surgery (n=74)
 Clinical or surgeon reason (n=11)
 No reason given (n=17)
 Other (n=6)
Did not want to consent (n=284)
 Did not want to be randomised (n=100)
 Personal reasons (n=81)
 Not interested (n=52)
 No reasons given (n=49)
 Other (n=2)
Other (n=21)
 Withdrawn post-consent, pre-randomisation (n=9)
 Not randomised after consent (n=12)

**Randomised (n=394)**

**Allocated to simple dressing (n=133; before wound closure n=66, after wound closure n=67)**

**Allocated to glue-as-a-dressing (n=129; before wound closure n=64, after wound closure n=65)**

**Allocated to 'no dressing' (n=132; before wound closure n=66, after wound closure n=66)**

**Withdrawal pre-surgery (n=0)**
**Protocol deviations (n=13)**
(3 & 10 were non-adherent to dressing allocation & timing of allocation respectively)

**Withdrawal pre-surgery (n=1)**
**Protocol deviations (n=17)**
(8 & 9 were non-adherent to dressing allocation & timing of allocation respectively)

**Withdrawal pre-surgery (n=0)**
**Protocol deviations (n=31)**
(20 & 11 were non-adherent to dressing allocation & timing of allocation respectively)

**Withdrawal post-surgery (n=6)**

**Withdrawal post-surgery (n=6)**

**Withdrawal post-surgery (n=4)**

**Numbers with follow up data**
Wound questionnaires* at 4 days (n=118 & 118 respectively)
EQ-5D at 15 days & 4-8 weeks (n=90 & n=84 respectively)
Observer & participant WHQ at 4-8 wks (n=93 & n=84 respectively)
In-hospital & reference SSI

**Numbers with follow up data**
Wound questionnaires* at 4 days (n=121 & 119 respectively)
EQ-5D at 15 days & 4-8 weeks (n=87 & n=78 respectively)
Observer & participant WHQ at 4-8 wks (n=92 & n=85 respectively)
In-hospital & reference SSI

**Numbers with follow up data**
Wound questionnaires* at 4 days (n=119 & 118 respectively)
EQ-5D at 15 days & 4-8 weeks (n=92 & n=80 respectively)
Observer & participant WHQ at 4-8 wks (n=101 & n=87 respectively)
In-hospital & reference SSI

Withdrawal pre-surgery as surgery cancelled. Withdrawals post-surgery: participant preference (n=9), death (n=2), randomisation failed in theatre (n=2), clinician chose to withdraw participant (n=2), and one participant required emergency re-operation. WHQ: Wound Healing Questionnaire, SSI: Surgical Site Infection

**Figure 2** Consort flow diagram of participants in the Bluebelle study. *Wound Management Questionnaire and Wound Experience Questionnaire.

**Table 1** Outcomes related to the feasibility of identifying and recruiting patients

| | NBT: general surgery | NBT: obstetric surgery | UHBham: general surgery | UHBris: general surgery | WORC: general surgery | Total |
|---|---|---|---|---|---|---|
| No. months open* | 7 | 7 | 9 | 9 | 4 | 36 |
| No. potentially eligible recorded/month (median, IQR) | 14 (3.0–25.0) | 27 (25.0–48.0) | 71 (57.0–80.0) | 21 (13.0–25.0) | 10 (4.5–13.0) | 142 (57.0–152.0) |
| No. potentially eligible recorded by staff | 96 | 230 | 558 | 196 | 35 | 1115 |
| No. (%) potentially eligible confirmed eligible | 90 (93.8) | 205 (89.1) | 469 (84.1) | 154 (78.6) | 34 (97.1) | 952 (85.4) |
| No. (%) of eligible who were approached | 87 (96.7) | 126 (61.5) | 317 (67.6) | 136 (88.3) | 33 (97.1) | 699 (73.4) |
| No. (%) of eligible approached and consented† | 65 (74.7) | 81 (64.3) | 120 (37.9) | 127 (93.4) | 22 (66.7) | 415 (59.4) |

*Nearest whole month.
†Not all consented patients were finally randomised.
NBT, North Bristol NHS Trust; UHBham, University Hospitals Birmingham NHS Foundation Trust; UHBris, University Hospitals Bristol NHS Foundation Trust; WORC, Worcestershire Acute Hospitals NHS Trust.

well although other follow-up assessments of SSI questionnaires were less satisfactory. Completeness of follow-up, however, was not the focus of the pilot study (foci were recruitment and adherence). It is expected that a future trial would combat these challenges using a complementary armamentarium of measures to enhance follow-up (reminders, text messages, telephone follow-up, etc).

Many previous RCTs have examined interventions to reduce SSI, although the quality and conduct of most studies is low and there is a lack of strategic feasibility work.[8–10] The Bluebelle study has addressed many of the key issues. Importantly, it demonstrates that a large, rigorous RCT could be done. In the participating centres there was, however, variation in rates of randomisation (37.9%–93.4%). Some of this variation is likely to be explained by the different approaches used to approaching and screening patients between hospitals. It may also reflect how the study was communicated by individuals at different centres. In a main trial, it is expected that training for recruitment and materials used to optimise recruitment will be available based on lessons learnt in this pilot. It is likely that a main two-group trial would address whether 'no dressings' are non-inferior to a simple dressing in terms of SSI; this is the comparison is of greatest value to the National Health Service.[18] A main trial with three groups would be more efficient than a separate trial to test the superiority of 'glue-as-a-dressing' to simple dressings. Although basic/simple dressings are inexpensive, they are used in very high volumes. Evidence that a 'no dressing' strategy is non-inferior may result in significant savings for the health service. However, providing this evidence would likely require likely a very large trial (>10 000 participants) to exclude the possibility of a small increase in the SSI rate in the no dressing group compared with the basic dressing group. Such a large trial would require an efficient design with electronic data capture and a well organised multidisciplinary clinical and academic team including patient partners. Since the conception of the Bluebelle study, there has been growing use of negative pressure wound therapy on primary wounds to reduce SSI. There is also increasing use of advanced dressings (with interactive properties). While these are of interest to the field, the focus of the proposed main Bluebelle trial is to establish whether 'no dressing' is non-inferior to standard dressings and to gain data to support the use of 'glue-as-a-dressing' on a primary surgical wound.

In the Bluebelle pilot RCT, there were contributions from surgical trainees as part of surgical research collaboratives. As observed in other studies, these collaboratives helped the trial to recruit to time and target.[20 21] Trainees were also involved in the study design (two trainees were grant coapplicants) and led and contributed to substudies. The involvement of surgical trainees in high-quality trials means that they can gain a research apprenticeship. This will equip their consultant practice with skills to engage in establishing evidence and implementing it as the results of trials become available. There were also complexities of working with surgical trainees, relating to the numbers of people involved and occasional confusion over responsibilities. Centres were required to set up additional processes to streamline communication between the teams and trainees. It is recommended that major trials involving trainee collaboratives consider budgeting for additional administrative support to allow coordination of the efforts of the large numbers of people involved.

Although the study recruited to time and target, there were limitations with the response rates to follow up

**Table 2** Demographics and clinical details of randomised participants by group

| | Simple dressing n=131 | Glue as-a-dressing n=126 | 'No dressing' n=131 | Total n=388 |
|---|---|---|---|---|
| Median age in years (IQR) | 55 (35.9–65.3) | 48 (32.3–66.2) | 53 (36.4–68.2) | 52 (34.7–66.9) |
| Female gender (%) | 80/131 (61.1) | 75/126 (59.5) | 72/131 (55.0) | 227/388 (58.5) |
| Median BMI (IQR)* | 28 (24.5–31.8) | 27 (24.2–32.0) | 28 (24.6–31.0) | 28 (24.3–31.6) |
| Ethnicity (%) white | 120/128 (93.8) | 105/119 (88.2) | 116/127 (91.3) | 341/374 (91.2) |
| Smoking history (%) | | | | |
| Current smoker | 16/131 (12.2) | 22/125 (17.6) | 22/130 (16.9) | 60/386 (15.5) |
| Ex-smoker >1 month | 53/131 (40.1) | 36/125 (28.8) | 47/130 (36.2) | 136/386 (35.2) |
| Current steroids, PO/IV/IM (%) | 15/131 (11.5) | 4/126 (3.2) | 6/131 (4.6) | 25/388 (6.4) |
| Diabetes, any type (%) | 11/130 (8.5) | 10/126 (7.9) | 8/130 (6.2) | 29/386 (7.5) |
| ASA class (%) | | | | |
| 1: Healthy, no medical problems | 43/128 (33.6) | 51/125 (40.8) | 40/131 (30.5) | 134/384 (34.9) |
| 2: Mild systemic disease | 72/128 (56.3) | 58/125 (46.4) | 73/131 (55.7) | 203/384 (52.9) |
| 3/4: Severe systemic disease | 13/128 (10.2) | 16/125 (12.8) | 18/131 (13.7) | 47/384 (12.2) |
| Wound closure (wounds/patients) | | | | |
| Sutures | 240/121 (95.3) | 240/117 (95.1) | 229/117 (90.7) | 709/355 (93.7) |
| Clips | 14/10 (9.9) | 13/6 (6.1) | 16/12 (11.5) | 43/28 (9.2) |
| Steristrips | 20/9 (7.1) | 1/1 (0.8) | 7/5 (3.8) | 28/15 (4.0) |
| Glue (not planned) | 4/2 (2.0) | 2/2 (2.0) | 4/2 (1.9) | 10/6 (2.0) |
| Total no of wounds | 278 | 256 | 256 | 790 |
| Prophylactic antibiotics (%) | 101/129 (78.3) | 99/126 (78.6) | 96/130 (73.8) | 296/385 (76.9) |
| Infection risk of surgery (%)† | | | | |
| Clean | 46/131 (35.1) | 49/126 (38.9) | 44/131 (33.6) | 139/388 (35.8) |
| Clean-contaminated | 81/131 (61.8) | 72/126 (57.1) | 81/131 (61.8) | 234/388 (60.3) |
| Contaminated/dirty | 4/131 (3.1) | 5/126 (4.0) | 6/131 (4.6) | 15/388 (3.9) |

*Four missing data (simple, glue-as-a-dressing, 'no dressing', 2, 1, 1, respectively), elsewhere when a cell denominator is different to the number in a column header, the difference arises because of missing data for that variable.
†Classified by type and urgency of surgery.
ASA, American Society of Anesthesia;BMI, body mass index; IM, intramuscular;IV, intravenous; PO, per oral.

assessments made by post. The logistics of obtaining the data were complex in this pilot study with three assessments being made (a patient-completed SSI assessment;

**Table 3** Potential trial primary outcome by group

| | Simple dressing n=131 | Glue as-a-dressing n=126 | 'No dressing' n=131 | Total n=388 |
|---|---|---|---|---|
| SSI (%)* | | | | |
| 4–8 week reference | | | | |
| None | 80/97 (82.5) | 83/98 (84.7) | 90/107 (84.1) | 253/302 (83.8) |
| Superficial | 14/97 (14.4) | 14/98 (14.3) | 17/107 (15.9) | 45/302 (14.9) |
| Deep | 3/97 (3.1) | 0/98 (0.0) | 0/107 (0.0) | 3/302 (1.0) |
| Organ/space | 0/97 (0.0) | 1/98 (1.0) | 0/107 (0.0) | 1/302 (0.3) |
| Overall | 17/92 (18.5) | 16/90 (17.8) | 18/99 (18.2) | 51/281 (18.1) |

*When the cell denominator is different to number in column header, the difference arises because of missing data for that variable.
SSI, surgical site infection.

an observer-completed SSI assessment and an independent face-to-face reference SSI assessment) and this required two members of staff. In a main trial, a single assessment would be required. It would also aim follow-up processes (scheduling of despatch and generation of questionnaires, etc) to be largely automated and for assessments to be conducted electronically (manual processes were used in this pilot RCT). It is, therefore, believed that it is possible to improve the response rate substantially and we have recommended to the funder that a future main trial be required to demonstrate a high response rate in an internal pilot phase. In the main trial, it may also be possible to supplement questionnaire data about SSIs with wound photography. Further work is ongoing exploring this.

In summary, this pilot RCT has informed the feasibility, design and likely conduct of a future main trial of different dressing strategies, including 'no dressing'.[16] A future three group trial could jointly address the hypotheses

**Table 4** Questionnaire response rates for SSI assessments, wound experience and management questionnaires and EQ-5D-5L by group and overall

| | Simple dressing n=131 (%) | Glue as-a-dressing n=126 (%) | 'No dressing' n=131 (%) | Total n=388 (%) |
|---|---|---|---|---|
| SSI reference assessment | 97/127 (76.4) | 98/122 (80.3) | 107/128 (83.6) | 302/377 (80) |
| Patient reported SSI assessment (WHQ) | 84/127 (66.1) | 85/122 (69.7) | 87/129 (67.4) | 256/378 (68) |
| Observer reported SSI assessment (WHQ) | 93/127 (73.2) | 92/122 (75.4) | 101/128 (78.9) | 286/377 (76) |
| Wound questionnaires | | | | |
| Experience | 118/131 (90.1) | 119/125 (95.2) | 118/129 (91.5) | 355/385 (92.2) |
| Management | 118/131 (90.1) | 121/125 (96.8) | 119/129 (92.2) | 358/385 (93.0) |
| EQ-5D-5L | | | | |
| Baseline | 128/131 (97.7) | 126/126 (100) | 131/131 (100) | 385/388 (99.2) |
| 15 days | 90/128 (70.3) | 87/125 (69.6) | 92/129 (71.3) | 269/383 (70.4) |
| 4–8 weeks | 84/127 (66.1) | 78/122 (63.9) | 80/128 (62.5) | 242/377 (64.2) |

EQ-5D-5L, EuroQoL-five-dimension-five-level; SSI, surgical site infection; WHQ, Wound Healing Questionnaire.

that: (1) 'glue-as-a-dressing' reduces the risk of SSI compared with 'simple dressing' (superiority of glue-as-a-dressing) and (2) 'no dressing' does not increase the risk of SSI (non-inferiority of 'no dressing'). In such a trial, it is proposed that the primary outcome should be a combination of information about SSI collected at discharge (as in this study) and SSI ascertained by the patient-reported questionnaire (WHQ), providing that a better response rate can be obtained and a cut-off score on WHQ can be established to define SSI. A conventional 'reference' SSI assessment would be impracticable as the primary outcome in a main trial because of the high cost of face-to-face assessments. In view of the observed rates of SSI in this pilot RCT and other studies, such a trial will need to be sizeable (>10 000 patients) to confidently exclude true differences in SSI rate. Another issue to consider for a main trial is the best time to disclose dressing allocation (before or after wound closure). It is concluded that the pilot RCT and feasibility work undertaken within the Bluebelle study has been valuable to inform surgical RCT design. This approach is recommended for other clinical questions with challenges in recruitment and outcome assessment before embarking on a main trial.

**Contributors** The Bluebelle study group consists of the co-investigators and other members. Full details of individuals contributions are as follows: Co-investigators: Mark Woodward: contributing experience of not using dressings on surgical wounds in children who have had abdominal surgery. Nicky J Welton: responsible for conducting a value for information analysis about a full-scale trial. Andrew D Torrance: recruited patients for the qualitative interviews and contributed to development of the SSI measure. Leila Rooshenas: responsible for design and delivery of qualitative studies in the pilot trial, to inform trial design and test various aspects of feasibility. Chris Rogers: responsibile for estimating the target sample size and planning the quantitative analyses. Barnaby C Reeves: responsible for the design and methods of the Bluebelle feasibility study. Anne Pullyblank: responsible for set up and delivery in one participating centre of the study in general surgery. Thomas D Pinkney: responsible for one participating centre recruiting patients having abdominal surgery and contributed to the overall study design and development of the WHQ. Jonathan M Mathers: responsibile for co-design and delivery of the qualitative studies in the pilot trial. Richard Lovegrove: responsibile for one participating centre recruiting patients having abdominal surgery. Robert J Longman: responsibile for one participating centre recruiting patients having abdominal surgery. Rachael Gooberman-Hill: responsibility for patient and public involvement. Jo C Dumville: updated the Cochrane review of wound dressings to consider the use of tissue adhesives as a dressing. Tim Draycott: responsibility for one participating centre recruiting women having caesarean section. Jenny L Donovan: responsibility for supervising qualitative research. Joanna Coast: responsibility for supervising health economic aspects of the pilot trial. Melanie J Calvert: advising and supporting development of the WHQ and other outcome measures, the feasibility study and pilot trial design. Jane M Blazeby: Chief investigator responsible for overall conception and the design of the Bluebelle feasibility study. Natalie S Blencowe: led the survey of wound dressings and contributed to the design of the pilot trial, the development of WHQ and other outcome measures. Lazaros Andronis: responsibility for carrying out health economic aspects of the pilot trial. Other study group members: Dimitrios Siassakos: responsible for implementation of the study protocol in one centre. Caroline Pope: assisted with setting up and managing the trial. Tom Milne: researched definitions of simple, complex and no dressings under supervision. Contributed to the development of the WHQ. Christel McMullan: responsible for qualitative data collection /analysis in the pilot trial. Rhiannon Macefield: led the development and validation of the Wound Healing Questionnaire. Rosie Harris: responsible for preparing the pilot RCT statistical analysis plan. Lucy Ellis: prepared the first protocol for the pilot trial. Daisy Elliott: led the development of other study outcome measures. Madeleine Clout: assisted in managing the pilot trial and member of the writing group for this manuscript. Benjamin E Byrne: helped set up one participating centre including designing training materials and improving study design and processes. Kate Ashton: managed the pilot trial, including site initiation visits and the second protocol amendment.The following members identified and recruited patients and contributed to study delivery in local trust: Benjamin R Waterhouse, Sean Strong, William Seligman, Lloyd Rickard, Samir Pathak, Anwar Owais, Jamie O'Callaghan, Stephen O'Brien, Dmitri Nepogodiev, Khaldoun Nadi, Charlotte Murkin, Tonia Munder, David Messenger, Matthew Mason, Morwena Marshall, Jessica Lloyd, Jeffrey Lim, Kathryn Lee, Vijay Korwar, Daniel Hughes, George Hill, Mohammed Hamdan, Hannah Gould Brown, James Glasbey, Caroline Fryer, Simon Davey, David Cotton, Oliver D Brown, Katarzyna D Bera, Joanne Bennett, Richard Bamford, Danya Bakhbakhi, Muhammad Atif, Elizabeth Armstrong, Piriyankan Ananthavarathan. The following nurses and administrative staff contributed to study delivery and management within the local research teams: Rebecca Houlihan, Joanna Nicklin, Louise Flintoff, Jo Chambers, Karen Bobruk (University Hospitals Bristol NHS Foundation Trust). Helen Cheshire, Suriya Kirkpatrick, Louise Solomon, Alice Jarvie, Cathy Winter, Clementine Skilton, Susan Hughes, Michelle Mayer, Jade Knowlden, Mary Alvarez, Sherrie Villis (North Bristol NHS Trust). Kelly Hollier, Natalie Jackson, Victoria Hardy, David Tyrell, Sharon Garner,

Arlo Whitehouse (University Hospitals Birmingham NHS Foundation Trust). Caroline Alton, Jessica Thrush, Julie Wollaston (Worcestershire Acute Hospitals NHS Trust). Katrina Hurley (Bristol Dental Hospital, Bristol). David Hutton, Helen Talbot (Clinical Trials and Evaluation Unit, Department of Translational Health Sciences, Bristol Medical School, Bristol). Laura Magill (University of Birmingham Clinical Trials Unit, Birmingham). Trudie Young (University of Glamorgan, Glamorgan). Katy Chalmers, Barry Main (Department of Population Health Sciences, Bristol Medical School, Bristol).

**Funding** The Bluebelle study was funded by the National Institute for Health Research HTA Programme (project number 12/200/04) and supported by the MRC ConDuCT-II Hub for Trials Methodology Research (Collaboration and innovation for Difficult and Complex randomised controlled Trials In Invasive procedures - MR/K025643/1). JLD and JB are NIHR senior investigators. JB is partially funded by the NIHR Biomedical Research Centre at University Hospitals Bristol NHS Foundation Trust and University of Bristol. MJC is partly funded by the NIHR Birmingham Biomedical Research Centre and the NIHR Surgical Reconstruction and Microbiology Research Centre at the University Hospitals Birmingham NHS Foundation Trust and the University of Birmingham. This study was designed and delivered in collaboration with the Clinical Trials and Evaluation Unit (CTEU), a UKCRC registered clinical trials unit which, as part of the Bristol Trials Centre, is in receipt of National Institute for Health Research CTU support funding.

**Disclaimer** The funders had no role in the design of the study, the collection and analyses of the data, the interpretation of the findings or in drafting this manuscript.

**Competing interests** None declared.

**Patient consent for publication** Not required.

**Ethics approval** Full research ethics approval was obtained from the Frenchay Research Ethics Committee on the 24th February 2015 (REC reference 15/SW/0008).

**Provenance and peer review** Not commissioned; externally peer reviewed.

**Data availability statement** Data are available on reasonable request. Requests can be made to bluebelle_trial@bristol.ac.uk. Anonymised individual participant data is available for secondary research, conditional on assurance that the proposed use of the data is compliant with the MRC Policy on Data Sharing regarding scientific quality, ethical requirements and value for money.

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
