## [Reviewer comments · BMJ Open]

ARTICLE DETAILS

TITLE (PROVISIONAL)	The Bluebelle pilot randomised controlled trial of three wound dressing strategies to reduce surgical site infection in primary surgical wounds
AUTHORS	Blazeby, Jane

VERSION 1 – REVIEW

REVIEWER	Prof Ian Chetter University of Hull, UK
REVIEW RETURNED	23-Apr-2019

GENERAL COMMENTS	Abstract Clear and concise apart from 699 / 1115 were eligible and approached is reported as 73.4% - should this be 62.7%? Introduction Reports 4.5 million hospital admissions for surgery annually in England – references a RCS document, which derives this data from Hospital episode statistics from 2009/10 i.e. 10 years ago. Could the reference be updated please Reports SSI rate following Caesarian section is “high”. It would be useful to know the precise figures from the literature In the final sentence of the introduction it states “The aim of the pilot RCT was to establish whether it would be feasible to carry out a large definitive RCT”. It would be helpful to include in this “aims” section the specific feasibility outcomes you aimed to assess. Methods Whilst I appreciate the protocol has been published elsewhere I think it important to include here more detail regarding the randomisation process (independent, online etc) and how this process required a second log in for patients randomised to “reveal dressing after wound closure”, as this may have influenced difference in adherence rates. The section “Additional wound assessments” is not reported in the results. Either remove this section from methods or include an appropriate results section. Results The first line states “1115 patients were screened; 699 (73.4%) were eligible and approached”. Should this be 62.7%? If so also needs addressing in abstract Second line states “415 consented to take part; 394 were randomised”. Please include percentages after figures. The SSI rate is reported as 18.1%, slightly lower than might be expected (i.e. in ROSSINI trial it was 25%). Was this reduction across all surgeries or perhaps explained by lower SSI rate following CS? Was there any difference in SSI rates between GS units? The authors should perhaps consider including SSI rates for CS and individual units Discussion The first paragraph is a good summary of feasibility outcomes but
---

	could be improved. I would suggest discussing the variability between units in the percentage of eligible patients who were approached and consented. This varied from 37.9% to 93.4%, and was lowest in the busiest unit. Discussion should probably include potential reasons, and the impact of interventions to address this.
--	---

REVIEWER	Professor Karen Ousey University of Huddersfield Institute of Skin Integrity and Infection Prevention UK
REVIEW RETURNED	24-May-2019

GENERAL COMMENTS	Thank you for your paper. The paper presents the results of the feasibility study in relation to a potential future RCT. I have a few comments for you to consider: Please define a 'simple wound dressing' - there are a range of dressings that may fall into this category and many clinicians use this term to mean different things. In the limitations section this sentence is unclear: Only 67% of participants completing the SSI questionnaire which will need to be addressed in a main trial - please reword Please consider rewording this: This is especially a problem in abdominal surgery and high-risk settings where rates of SSI may reach 25 to This is especially problematic in abdominal surgery and high-risk settings where rates of SSI may reach 25%. Please identify where this research can be located: Caesarean section is another procedure which carries a high rate of SSI - also what is the percentage of C section infections? Line 31: Cochrane review of dressing strategies, which also reviewed evidence when wounds are left uncovered...this sentence is confusing and lacks clarity e.g. reviewed which evidence for what? Please clarify Line 42-43 - UK Institute of Health - is this the Institute for Health and Social Care? Page 8, line 30: please review sentence: participant has having had an SSI or not. - I think this should be participant as having had an SSI or not. Page 9 - line 52: Please review this sentence: The main trial will continue to include patients are all stages of the work. This may be my naivety but please can you clarify what a 'skin transfer' is?
--

REVIEWER	Prof MA Boermeester Amsterdam UMC Advisory board of and/or speaker for Johnson&Johnson, KCI/Acelity, Smith&Nephew
REVIEW RETURNED	10-Jun-2019

GENERAL COMMENTS	Systematic reviews - of studies of low quality - found no evidence to suggest that covering surgical wounds with dressings reduces the risk of a SSI, or that any particular wound dressing is more effective than another in reducing scarring, controlling pain, promoting patient acceptability or ease of dressing removal. Present study is in two parts (Phase A and Phase B). Phase A is completed and comprised interviews with surgeons, nurses/midwives and participants to investigate the current use of dressings and views about not using dressings, a multicentre survey to establish current practice relating
---

	to dressings. Phase B is a pilot RCT. Comments: As Phase A is completed and practical wound management and patient symptom experience measures have come forward from this Phase needed for the next phase, results from Phase B need to be included in present study protocol. It is unreal to ignore the fact that Phase A is already completed and - if correct - knowledges gains have been accomplished. And the paper states that Phase A 'will also be a further platform to refine the SSI measure and it will explore methods for blinding outcome assessors.' We need that before the start of Phase B, thus in present protocol paper. What exactly have the authors learned from Phase A results? The choice of wound dressings (simple, glue, none) for the pilot RCT is unclear. Why are advanced dressings left out. The data of previous trials comparing simple dressing to advanced dressings are of low quality; we cannot exclude advanced dressings. Moreover, in almost all types of surgical procedures (WHO Prevention of SSI guideline systematic review appendix), closed-incision prophylactic negative pressure wound therapy as postoperative 'dressing' is beneficial compared to standard dressing with respect to SSI outcome. Thus, you can perform a methodologically correct trial but if the treatment choices or comparisons of arms in your trial are not correct or lag behind modern views of wound care the trial results are meaningless. Comparing ineffective vs. ineffective wound dressings will probably render no difference in outcome but thereby does not reflect the best possible outcome. That is the way the (Cochrane) systematic reviews have interpreted trial data comparing dressings, coming to that very conclusion of 'no difference in outcome' whereas the correct interpretation is 'we are not there yet in terms of best possible outcome'. How good is wound assessment through judgment of wound photographs, in the sense of interrater reliability and in comparison to live wound assessment?
--	--

REVIEWER	Irina Chisster St George's University of London
REVIEW RETURNED	13-Aug-2019

GENERAL COMMENTS	This is a very good manuscript. I particularly commend the involvement of professional statisticians in the study design, analysis plan and hence a very clear description of statistics even for this initial pilot. One minor observation regards the Table 1: I am not sure of the relevance of the summary under the column indicating the Total. the statisticians may revise this. Other than that, I cannot comment on the clinical side of the research as this is beyond my area of expertise.
--

VERSION 1 – AUTHOR RESPONSE

Comments from reviewer 1

1. Clear and concise apart from 699 / 1115 were eligible and approached is reported as 73.4% - should this be 62.7%?

Response: The apparent discrepancy arises from the 21 labelled as "other" in the CONSORT diagram. These were participants who consented but who were not randomised because the trial stopped, i.e. they were in the process of being recruited and (we think) had consented, giving a total

consented of 394+21=415. The number eligible and approached is correct: 1115-163 (ineligibles)-253 (not approached) = 699, of whom 284 did not consent.

2. Introduction

Reports 4.5 million hospital admissions for surgery annually in England – references a RCS document, which derives this data from Hospital episode statistics from 2009/10 i.e. 10 years ago. Could the reference be updated please.

Response

We have updated this reference (no. 1).

3. Reports SSI rate following Caesarean section is “high”. It would be useful to know the precise figures from the literature

Response

We have added a relevant reference to support this statement (reference no. 7)

4. In the final sentence of the introduction it states, “The aim of the pilot RCT was to establish whether it would be feasible to carry out a large definitive RCT”. It would be helpful to include in this “aims” section the specific feasibility outcomes you aimed to assess.

Response

We have updated the aims to describe the feasibility outcomes.

“Specific objectives were to establish if it was possible to recruit and randomise, to assess the acceptability of, and adherence to, the trial interventions, to examine the feasibility of collecting follow up data, and to establish the measurement properties of the SSI Wound Healing Questionnaire.”

5. Methods

Whilst I appreciate the protocol has been published elsewhere I think it important to include here more detail regarding the randomisation process (independent, online etc) and how this process required a second log in for patients randomised to “reveal dressing after wound closure”, as this may have influenced difference in adherence rates.

Response

We updated this section to provide more details as requested

“The randomisation sequences were generated by computer in advance of starting to recruit. Allocation was concealed until participant’s eligibility and consent were documented and it was obtained via the internet. Depending on the randomisation result, the dressing allocation was either disclosed immediately, or the user was advised to log back on to the website after the wound had been closed. At the second log on the user was asked to record the timing of wound closure, then the allocation was disclosed.”

6. The section “Additional wound assessments” is not reported in the results. Either remove this section from methods or include an appropriate results section.

Response

We apologise for this error. We have removed this section from the method section page 9.

7. Results

The first line states "1115 patients were screened; 699 (73.4%) were eligible and approached". Should this be 62.7%? If so also needs addressing in abstract

Please see our response to the above reviewer point no. 1

8. Second line states "415 consented to take part; 394 were randomised". Please include percentages after figures.

Response

We have updated the text as below

"415 (37.2%) consented to take part; 394 (35.5%) were randomised".

9. The SSI rate is reported as 18.1%, slightly lower than might be expected (i.e. in ROSSINI trial it was 25%). Was this reduction across all surgeries or perhaps explained by lower SSI rate following CS? Was there any difference in SSI rates between GS units? The authors should perhaps consider including SSI rates for CS and individual units

We agree that the SSI rate is lower than that reported in the ROSSINI trial. We agree that this reduction is likely because our patient population includes all surgeries and not just patients having a laparotomy as per the ROSSINI study. We have not looked at SSI rates between individual units or made specific comparisons. This was a pilot study and the overall rate is therefore imprecise. The future main trial will be considering these issues.

10. Discussion

The first paragraph is a good summary of feasibility outcomes but could be improved. I would suggest discussing the variability between units in the percentage of eligible patients who were approached and consented. This varied from 37.9% to 93.4% and was lowest in the busiest unit. Discussion should probably include potential reasons, and the impact of interventions to address this.

Response

Thank you for this feedback. We have updated the first paragraph of the discussion as below to consider these issues.

"In the participating centres there was, however, variation in rates of randomisation (37.9% to 93.4). Some of this variation is likely to be explained by the different approaches used to approaching and screening patients between hospitals. It may also reflect how the study was communicated by individuals at different centres. In a main trial it is expected that training for recruitment and materials used to optimise recruitment will be available based on lessons learnt in this pilot."

Reviewer: 2

Reviewer Name: Professor Karen Ousey

Please leave your comments for the authors below

Thank you for your paper. The paper presents the results of the feasibility study in relation to a

potential future RCT. I have a few comments for you to consider:

1. Please define a 'simple wound dressing' - there are a range of dressings that may fall into this category and many clinicians use this term to mean different things.

Response

We apologise for not making this clearer. There is a full definition of a 'simple wound dressing' in the published protocol. We have edited the methods section here to draw attention to that.

"Full details of the interventions are described in the protocol.¹⁵"

2. In the limitations section this sentence is unclear: Only 67% of participants completing the SSI questionnaire which will need to be addressed in a main trial - please reword

Response

We have reworded this sentence

3. Please consider rewording this: This is especially a problem in abdominal surgery and high-risk settings where rates of SSI may reach 25 to. This is especially problematic in abdominal surgery and high-risk settings where rates of SSI may reach 25%

Response

We have reworded this sentence as suggested.

"This is especially problematic in abdominal surgery and high-risk settings where rates of SSI may reach 25%".

4. Please identify where this research can be located: Caesarean section is another procedure which carries a high rate of SSI - also what is the percentage of C section infections?

Response

We have added reference no. 7 which shows the high rates of SSI following caesarean section.

5. Line 31: Cochrane review of dressing strategies, which also reviewed evidence when wounds are left uncovered...this sentence is confusing and lacks clarity e.g. reviewed which evidence for what? Please clarify

Response

We have updated this sentence to improve clarity

"A Cochrane review of RCTs examining different dressing strategies, which included studies of wounds without a dressing, was performed in 2011 and since updated.^{8,9}"

6. Line 42-43 - UK Institute of Health - is this the Institute for Health and Social Care?

Response

Thank you for highlighting this error. It is the UK National Institute of Health Research

We have corrected this sentence

7. Page 8, line 30: please review sentence: participant has having had an SSI or not. - I think this should be participant as having had an SSI or not.

Response

We thank you for highlighting this error. We have corrected this now.

8. Page 9 - line 52: Please review this sentence: The main trial will continue to include patients are all stages of the work.

Response

Thanks for highlighting this confusing sentence. It has been updated as below

“The main trial will continue to include patients throughout all of its stages (design, delivery, analyses, reporting and implementation).”

9. This may be my naivety, but please can you clarify what a 'skin transfer' is?

Response

A skin transfer is a temporary adherent “tattoo” that is placed on the skin. It wears off naturally after about 10 days. We have updated the text to clarify this in the methods section.

“Skin transfers (temporary adherent tattoos) were placed”

Reviewer: 3

Reviewer Name: Prof MA Boermeester

Institution and Country: Amsterdam UMC

Please state any competing interests or state ‘None declared’: Advisory board of and/or speaker for Johnson&Johnson, KCI/Acelity, Smith&Nephew

Please leave your comments for the authors below

Systematic reviews - of studies of low quality - found no evidence to suggest that covering surgical wounds with dressings reduces the risk of a SSI, or that any particular wound dressing is more effective than another in reducing scarring, controlling pain, promoting patient acceptability or ease of dressing removal. Present study is in two parts (Phase A and Phase B). Phase A is completed and comprised interviews with surgeons, nurses/midwives and participants to investigate the current use of dressings and views about not using dressings, a multicentre survey to establish current practice relating to dressings. Phase B is a pilot RCT.

Comments:

1. As Phase A is completed and practical wound management and patient symptom experience measures have come forward from this Phase needed for the next phase, results from Phase B need to be included in present study protocol. It is unreal to ignore the fact that Phase A is already completed and - if correct - knowledges gains have been accomplished. And the paper states that Phase A 'will also be a further platform to refine the SSI measure and it will explore methods for

blinding outcome assessors.' We need that before the start of Phase B, thus in present protocol paper. What exactly have the authors learned from Phase A results?

Response

The main findings from Phase A have been reported elsewhere. The introduction of this paper has been edited to describe the work from Phase A and how it informed Phase B. The publications from that work are all now cited

“The Bluebelle study, a programme of research including non-randomised feasibility projects (Phase A) and a pilot RCT (Phase B) was designed to inform the design of a main trial¹²⁻¹⁶. Phase A included interviews with key stakeholders to explore their views of dressings and a trial design¹², a survey of surgical wounds to examine current dressing practice¹³ and developmental work to design questionnaires to assess SSI¹⁴ and other aspects of wound management¹⁵. Here we report the pilot RCT¹⁶.

2. The choice of wound dressings (simple, glue, none) for the pilot RCT is unclear. Why are advanced dressings left out? The data of previous trials comparing simple dressing to advanced dressings are of low quality; we cannot exclude advanced dressings. Moreover, in almost all types of surgical procedures (WHO Prevention of SSI guideline systematic review appendix), closed-incision prophylactic negative pressure wound therapy as postoperative 'dressing' is beneficial compared to standard dressing with respect to SSI outcome. Thus, you can perform a methodologically correct trial but if the treatment choices or comparisons of arms in your trial are not correct or lag behind modern views of wound care the trial results are meaningless. Comparing ineffective vs. ineffective wound dressings will probably render no difference in outcome but thereby does not reflect the best possible outcome. That is the way the (Cochrane) systematic reviews have interpreted trial data comparing dressings, coming to that very conclusion of 'no difference in outcome' whereas the correct interpretation is 'we are not there yet in terms of best possible outcome'.

Response

We thank you for these pertinent comments. We agree that trials of negative pressure wound therapy and advanced dressings are important. These are currently being carried out in the UK. In the Bluebelle study, however, the primary focus was to examine whether a trial of 'no dressings' was possible. We have demonstrated it is possible for patients with primary surgical wounds. In addition a trial including 'glue as a dressing' is novel as there are few studies with these intervention.

We have updated the discussion to highlight the importance of other trials including advanced dressings and prophylactic negative pressure wound therapy.

“Since the conception of the Bluebelle study there has been growing use of negative pressure wound therapy on primary wounds to reduce SSI. There is also increasing use of advanced dressings (with interactive properties). Whilst these are of interest to the field, the focus of the proposed main Bluebelle trial is to establish whether 'no dressing' is non-inferior to standard dressings and to gain data to support the use of 'glue-as-a-dressing' on a primary surgical wound.”

3. How good is wound assessment through judgment of wound photographs, in the sense of interrater reliability and in comparison to live wound assessment?

Response

This is a very important question. In the Bluebelle study we were unable to examine these issues because of the regulatory challenges with undertaking wound photographs in the different hospitals. It was therefore not possible to make this comparison

The text in the discussion has been updated to explain this issue as below

“In the main trial it may also be possible to supplement questionnaire data about SSIs with wound photography. Further work is on-going exploring this.”

Reviewer: 4

Reviewer Name: Irina Chisster

Institution and Country: St George's University of London

Please state any competing interests or state ‘None declared’: None declared

Please leave your comments for the authors below

1. This is a very good manuscript. I particularly commend the involvement of professional statisticians in the study design, analysis plan and hence a very clear description of statistics even for this initial pilot.

One minor observation regards the Table 1: I am not sure of the relevance of the summary under the column indicating the Total. the statisticians may revise this.

Response

We understand that the table could be presented with or without the total column. We are keen to keep this in because some readers will just look at those figures and be less interested in the individual site contributions